# Real Time Intrarenal Pressure Control during Flexible Ureterorrenscopy Using a Vascular PressureWire: Pilot Study

**DOI:** 10.3390/jcm12010147

**Published:** 2022-12-24

**Authors:** Alba Sierra, Mariela Corrales, Merkourios Kolvatzis, Steeve Doizi, Olivier Traxer

**Affiliations:** 1Urology Department, Hospital Clínic de Barcelona, Universitat de Barcelona, Villarroel 170, 08036 Barcelona, Spain; 2Sorbonne University GRC Urolithiasis No. 20 Tenon Hospital, F-75020 Paris, France; 3Department of Urology AP-HP, Tenon Hospital, Sorbonne University, F-75020 Paris, France; 42nd Department of Urology, Papageorgiou General Hospital, Aristotle University of Thessaloniki, 54124 Thessaloniki, Greece

**Keywords:** endourology, intrapelvic pressure, intrarenal pressure, ureteroscopy

## Abstract

(1) Introduction: To evaluate the feasibility of measuring the intrapelvic pressure (IPP) during flexible ureterorenoscopy (f-URS) with a PressureWire and to optimize safety by assessing IPP during surgery. (2) Methods: Patients undergoing f-URS for different treatments were recruited. A PressureWire (0.014”, St. Jude Medical, Little Canada, MN, USA) was placed into the renal cavities to measure IPP. Gravity irrigation at 40 cmH_2_O over the patient and a hand-assisted irrigation system were used. Pressures were monitored in real time and recorded for analysis. Fluid balance and postoperative urinary tract infection (UTI) were documented. (3) Results: Twenty patients undergoing f-URS were included with successful IPP monitoring. The median baseline IPP was 13.6 (6.8–47.6) cmH_2_O. After the placement of the UAS, the median IPP was 17 (8–44.6) cmH_2_O. With irrigation pressure set at 40 cmH_2_O without forced irrigation, the median IPP was 34 (19–81.6) cmH_2_O. Median IPP during laser lithotripsy, with and without the use of on-demand forced irrigation, was 61.2 (27.2–149.5) cmH_2_O. The maximum pressure peaks recorded during forced irrigation ranged from 54.4 to 236.6 cmH_2_O. After the surgery, 3 patients (15%) presented UTI; 2 of them had a positive preoperative urine culture, previously treated, and a positive fluid balance observed after the surgery. (4) Conclusion: Based on our experience, continuous monitoring of IPP with a wire is easy to reproduce, effective, and safe. In addition, it allows us to identify and avoid high IPPs, which may affect surgery-related complications.

## 1. Introduction

The development of fibre optic technology, digital ureteroscopes, and novel laser techniques have allowed the downsizing of flexible ureteroscopes, allowing not only treatment but also the diagnosis of many upper urinary tract conditions, such as kidney stones, ureteral strictures, and low-risk upper urothelial tumours [1,2]. Nevertheless, an adequate irrigation flow is required to achieve and maintain good visualization during these procedures [3].

With the downsizing of ureteroscopes, the working channel is typically reduced to 3.6 Fh. In endoscopic procedures, visibility is crucial, and it depends largely on the balance between the inflow, based on the irrigation pressure system and the working channel size, and the irrigation outflow, which depends on scope size and its relationship with the ureteral access sheath (UAS) [4]. The intrapelvic pressure (IPP) reached during f-URS is a result of irrigation inflow and outflow [3]. The physiological IPP ranges from 0 to 5 cmH_2_O and the pyelo-venous backflow occurs at pressures of 40.8–47.6 cmH_2_O [5,6]. During f-URS, when a disbalance occurs, high levels of IPP may be reached intraoperatively, causing pyelo-venous, and pyelo-lymphatic backflow or even rupture of the collecting system, possibly leading to peri-renal hematoma or urosepsis [5,7,8]. Prior in vivo studies have reported pressures as high as 436.9 cmH_2_O during f-URS [9], massively exceeding the pressure of pyelovenous backflows.

Despite some clinical experiences [10] with the current endourology armamentarium, we are not able to measure real-time in vivo intrarenal pressure during endourological procedures. The aim of our study is to evaluate simultaneously the IPP values using a vascular PressureWire and avoid sudden pressure increases during different f-URS procedures.

## 2. Materials and Methods

### 2.1. Study Design

A prospective pilot study of consecutive patients undergoing f-URS for different treatments, including kidney stone disease, pyelo-ureteral junction syndrome (UPJ) and diagnosis/treatment for upper tract urothelial carcinoma (UTUC), was performed between March and April 2022

### 2.2. Method of IPP Measurement

The PressureWire (St. Jude Medical, Saint Paul, MN, USA) was used before by Doizi et al. for IPP monitoring [9]. This 0.014′′ wire is approved and routinely used by cardiologists to assess fractional flow reserve in coronary arteries. The distal 3 cm of the wire, where the digital sensor is positioned to measure pressure, is made of soft platinum, which is floppy, radiopaque, hydrophilic and allows for positioning without renal trauma. In the following 28 cm, the wire is made of a polytetrafluoroethylene coating and is flexible and hydrophilic. Wirelessly, the pressure signal is transmitted to a console (QUANTIEN system) that displays the pressure (Figure 1). Pressure is recorded every second. The pressure is measured in mmHg and the available range is from −30 to 300 mmHg (−40.8 to 407.9 cmH_2_O). Its accuracy is ±1 mmHg plus ± 1% (≤50 mmHg) ± 3% (>50 mmHg). Pressure values measured in mmHg were multiplied by 1.35951 to convert them in cmH_2_O.

### 2.3. Procedures

Perioperative antibiotic prophylaxis was administrated following the local protocol. All procedures were performed under general anaesthesia. Each procedure began with a cystoscopy and the placement of a hydrophilic guidewire in the renal pelvis under fluoroscopic guidance. A dual lumen ureteral catheter (Cook Medical, Bloomington, IN, USA) was then inserted and the PressureWire was placed in the renal pelvis for IPP measurements (Figure 2). Once the dual lumen catheter was removed, the f-URS was either passed directly over the hydrophilic guidewire or, when indicated, through a UAS inserted over the hydrophilic guidewire (Retrace 10/12 or 12/14, 35 cm, Coloplast, Humlebaek, Denmark). In some cases, PressureWire was placed into the UAS (Figure 3). Retrograde intrarenal surgery (RIRS) was performed using a flexible digital re-usable ureteroscope, the Flex—Xc (Karl Storz, Tuttlingen, Germany), with a constant 0.9% saline irrigation pressure (40 cmH_2_O) at ambient temperature and a manual pump (Traxerflow Dual Port, Rocamed, Monaco), allowing on-demand forced irrigation when a better view was required. All of the interventions performed by an experienced endourologist (OT). The assistant controlled the pressure during the entire surgery, ensuring good vision and trying not to exceed values above 60 cmH_2_O (Figure 4). When laser treatment was needed, a thulium fibre laser (SOLTIVE Premium, Olympus, Tokio, Japan or FIBERDUST, Quanta System, Samarate, Italy) was used. At the end of each surgery, we inserted a ureteral stent (Double J) for 7–10 days. Patients were followed in the postoperative period to identify any possible complications.

### 2.4. Data Collection

-Baseline IPP: recorded with only the PressureWire in place, prior to f-URS insertion and irrigation flow.-UAS IPP: Recorded when placing the UAS-Scope IPP: Recorded during the introduction of the flexible ureteroscope into the renal cavities and irrigation pressure set at 40 cmH2O without any forced irrigation.-Therapeutic period IPP: Once reaching a plateau for 30 s. In real time, the assistant responsible for forced irrigation was aware of IPP measurements.

In case of stone disease, patients underwent non-contrast-enhanced CT for stone volume, which was obtained with the formula of an ellipsoid (4/3 × π × radius length × radius width × radius depth). Median IPP values, peak pressures, and pressure patterns with and without the scope in the renal cavity were examined, as well as the influence of on-demand irrigation during the surgical procedure. The fluid balance (FB) was the difference between the saline irrigation volume used during the surgery and the volume in the vacuum at the end of the surgery. During the hospitalisation, postoperative complications were recorded. For statistical analysis, categorical variables were measured as percentages and numerical variables were expressed as medians (interquartile range (IQR)).

## 3. Results

Of the 20 patients included in this study, 55% (n = 11) were male and 45 (n = 9) female. The median age was 51 (19–79) years old. Placement of the PressureWire succeeded in all cases and IPP measurements were obtained in all cases (Table 1).

Two patients with UTUC, one for surveillance and the other one for endoscopic treatment, had baseline pressures of 15 cmH_2_O in both cases. Therapeutic IPP was 57 cmH_2_O. However, the maximum peak pressure recorder was 114.2 cmH_2_O.

One patient with pyelo-ureteral junction syndrome demonstrated a pressure two times higher than the baseline pressure after the administration of furosemide iv (1 mg).

f-URS was performed for stone lithotripsy in 85% of cases (n = 17). Four of them were pre-stented. The median stone burden was 864 (50–9000) mm^3^. Overall, 52% (n = 9) were calcium oxalate stones. The median baseline IPP was 13.6 (6.8–47.6) cmH_2_O. UAS was used in 14 patients (70%), mostly 10/12 Fr, according to the surgeon’s choice. After UAS placement, the median UAS IPP was 17 (8–44.6) cmH_2_O. During f-URS, with the endoscope in the renal cavity and irrigation pressure set at 40 cmH_2_O without any forced irrigation, the median IPP was 37.4 (19–81.6) cmH_2_O when UAS was used and 35.2 (21.8–64) cmH_2_O without UAS. We controlled the pressure simultaneously during all of the surgeries. When forced irrigation was used, immediate IPP changes were observed, according to the way in which the assistant used the irrigation system. The median IPP during therapeutic period with the use of on-demand forced irrigation was 61.2 (27.2–149.5) cmH_2_O. The maximum pressure peaks recorded during this period ranged from 54.4 to 238 cmH_2_O.

The median surgery time was 149.5 (60–256) min. Positive preoperative urine culture was detected in 25% (n = 5) patients, all of them with renal stones (Table 2). According to the antibiogram, antibiotherapy was started 3 days before the surgery in all cases. Overall, 15% (n = 3) of patients were diagnosed with a UTI after the procedure. The complication rate was low and mostly Clavien–Dindo grade I and II. There were no complications related to PressureWire placement.

## 4. Discussion

Pyelovenous backflow, which occurs at pressures of 40.8–47.6 cmH_2_O, is an event that most urologists try to avoid [5,6]. That is why an IPP around 40 cmH_2_O is recognised as an aspirational threshold and should be the goal during endourological procedures [2]. In our pilot study, although IPP was rigorously controlled, maintaining IPP around 40 cmH_2_O was not feasible to maintain good visualization. We target pressures as low as possible, achieving 61.2 cmH_2_O median IPP. In a recent systematic review, IPP at 40 cmH_2_O was also exceeded during ureterorenoscopic procedures, specially without UAS [10]. Additionally, if we consider high-power laser lithotripsy, moderate irrigation is needed for the laser to be safe, because if irrigation rates decrease, we can produce a significant temperature increase, potentially resulting in urothelial tissue injuries [11]. Understanding this fact is crucial when interpreting findings, since improving drainage may be preferable compared to decreasing irrigation pressure/flow.

Unlike prior in vivo human studies where a ureteral catheter or a nephrostomy tube were used [12,13], we placed a 0.014” PressureWire in renal cavities. This IPP method measurement was described previously by Doizi et al. [9]. This system offers several advantages: it can be used for endoscopic procedures with all scope brands, and as the wire is placed into renal cavities, we can control IPP throughout the the procedure, because in addition to working along the ureter to treat, e.g.., a ureteral stone, which is important, it can work up to the pyeloureteral junction [3,5]. However, its small size prevents us from using it as a safety wire, needing us to place both the PressureWire and a safety wire.

Regardless of the IPP measurement method, with gravity irrigation at 40 cmH_2_O, similar baseline IPPs were also reported in the literature, ranging from 23.8 to 57 cmH_2_O without UAS and 13.14 to 33.99 cmH_2_O with a 10/12 UAS [10]. In addition, the scope IPP without UAS was two to three times higher than baseline IPP, which demonstrates once again that higher IPP is achieved without UAS [4]. Concerning therapeutic IPP, no comparison can be performed with previous studies, since many parameters differ: f-URS model, gravity and forced irrigation, pre-stenting and use or not of UAS and its size.

Prior in vivo human studies have reported peak pressures above 400 cmH_2_O [9,10,13]. In our cohort, by means of simultaneous IPP control, we halved these values for a short period of time. By means of simultaneous IPP control, we can quicky react to decreased IPP, avoiding pathological kidney changes reported in the literature [14]. In this line, in the immediate follow-up, no urinary extravasation was identified. However, fluid absorption was noted in four patients. Fluid absorption during f-URS usually remains low, mainly due to the smaller instrument calibre and the small irrigation channel. Nevertheless, increasing the flow to maintain optimal visibility necessitates the use of high-pressure irrigation, thus increasing the risk of fluid extravasation. IPP is not the only parameter to consider during fluid absorption; urothelial damage and surgery length are also important. Cybulski et al. reported that there is approximately 1 mL of irrigation fluid absorbed per minute of URS time at 271.9 cmH_2_O [15].

Additionally, the procedure time is independently correlated with increased postoperative fever and SIRS rates [16]. There is probably a correlation between IPP and infectious complications such as UTI and sepsis during endourological procedures, as well as other factors such as patient age, stone size and type, and length of the surgery [10,17]. For instance, 15% of the patients in our series presented with postoperative UTI despite therapeutic IPP at 40 cmH_2_O, meaning that other factors may contribute to the development of infectious complications.

We are convinced that the next step to improve safety during intrarenal procedures will be IPP monitoring. In this line, the recently developed LithoVueTM Elite System (BostonScientific, Boston, MA, USA) might contribute to safety and will provide us with more information about intrarenal pressure during endourologic procedures. However, for now, this new device needs to be evaluated, since the post-market study recently started in July 2022. Moreover, as the pressure sensor is located on the scope’s tip, to measure IPP we will need the scope to be placed inside the kidney, while with the PressureWire we can control IPP throughout the procedure. In future research, it will be interesting to compare both methods of pressure monitoring.

## 5. Conclusions

In our experience, the use of the PressureWire for IPP measurement during therapeutic and diagnosis f-URS is simple, safe, reproducible, and independent of the f-URS procedure. Continuously monitoring the IPP in real time allows us to identify and avoid high IPPs, which may lead to surgery-related complications.

## Figures and Tables

**Figure 1 jcm-12-00147-f001:**
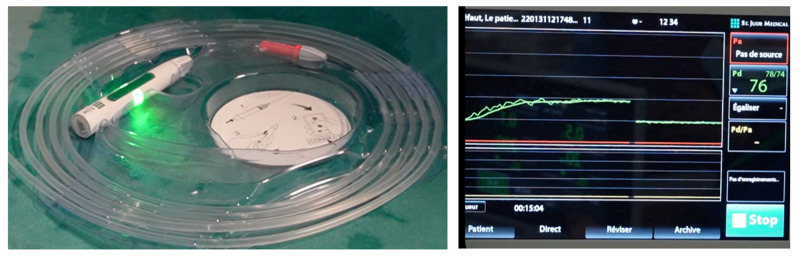
Wireless system. The PressureWire is activated by pressing a button (green light) and automatically connected wirelessly to a console (QUANTIEN system). The zeroing must be completed before the PressureWire placement, outside the patient. Once it is connected, it starts to simultaneously transmit the pressure signal to the screen.

**Figure 2 jcm-12-00147-f002:**
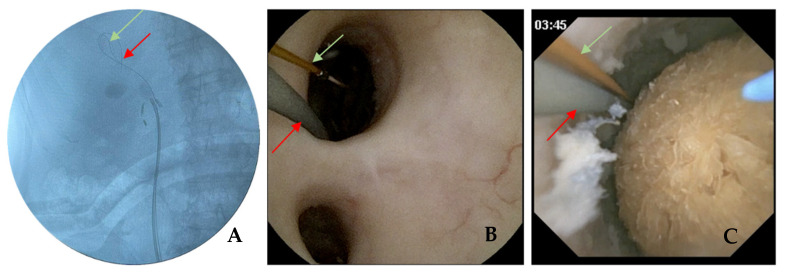
Pressure wire placement intro renal pelvis: (**A**) fluoroscopic image of PressureWire (green) and safety wire (red) in the renal pelvis. (**B**) Endoscopic vision of the renal pelvis with a PressureWire (green) and safety wire (red) going inside the upper calyx. (**C**) Endoscopic vision before starting lithotripsy of a dihydrate calcium oxalate stone with a safety wire (red), PressureWire (green) and fibre laser in the renal pelvis.

**Figure 3 jcm-12-00147-f003:**
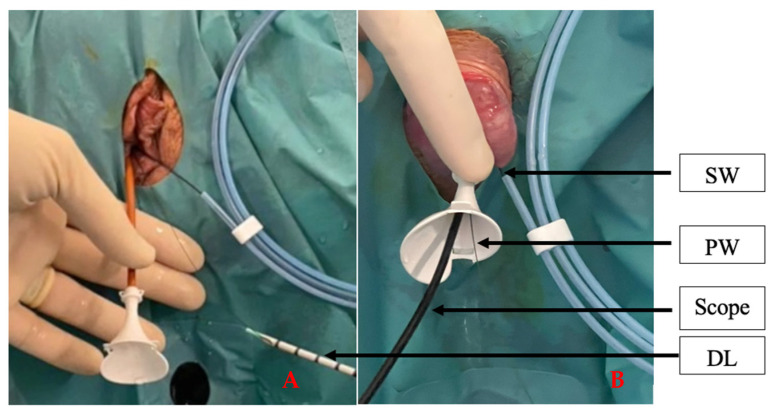
(**A**) Placement of PressureWire after the use of a dual lumen ureteral access catheter (Cook Medical, Germany). (**B**) PressureWire was placed through the UAS (Retrace 10/12, 35 cm, Coloplast, Denmark) with a digital reusable flexible ureterorenoscope (Flex-XC, 8.5Fh, Storz, Germany) inside. SW, safety wire. PW, PressureWire. DL, dual lumen.

**Figure 4 jcm-12-00147-f004:**
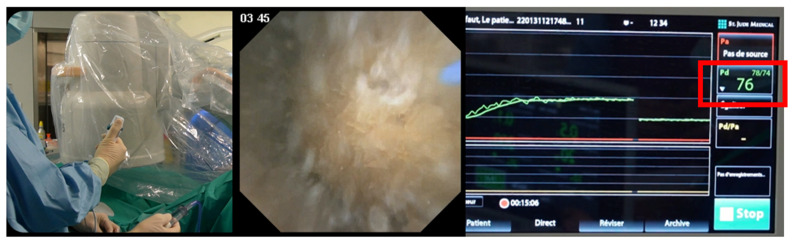
Pressure is simultaneously transmitted to the screen during endoscopic procedure. The pressure is measured in mmHg and the available range is from −30 to 300 mmHg. In this figure, during manual assisted irrigation (Traxerflow Dual Port, Rocamed, Monaco), we achieve IPP at 76 mmHg.

**Table 1 jcm-12-00147-t001:** Patient characteristics and intrapelvic pressures during flexible ureteroscopy.

Patient	Treatment	Stone Burden (mm^3^)	Stone Type	Pre-Stented	Fibre Size (µm)	UAS Size (Fr)	IPP (cmH_2_O)				
							Baseline	UAS	Scope	Therapeutic Period	Maximum Pressure Peak
1	Lithotripsy	9000	Cystine	Yes	200	10/12	47.6	44.6	40.8	54.4	163
2	Lithotripsy	7500	Infective	No	200	10/12	20.4	21.3	68	68	197
3	Lithotripsy	3190	Infective	No	200	10/12	16.3	20.4	27.2	68	176.7
4	Lithotripsy	864	Carbapatite	No	200	10/12	16.3	18.2	19	27.2	54.4
5	Pyeloureteral junction syndrome			No			13.6	30 (furosemide)			
6	Lithotripsy	3000	Cystine	Yes	272	10/12	13.6	15	81.6	136	238
7	Upper tract urinary tumour (diagnosis)			No			15	No	30	No	15
8	Lithotripsy	6000	OCD	Yes	200	10/12	20.4	27.2	61.2	149.5	236.6
9	Upper tract urinary tumour (treatment)			No	272	No	15	No	15	No	34
10	Lithotripsy	864	OCM	No	200	12/14	21.8	24.5	51.7	110.1	171.3
11	Lithotripsy	4000	OCD	No	200	10/12	13.6	13.6	31.3	54.4	99.2
12	Lithotripsy	5410	OCD	No	150	10/12	12.2	16.3	47.6	117	156.3
13	Lithotripsy	740	Mixed	No	200	10/12	12.2	13.6	20.4	70.7	122.4
14	Lithotripsy	270	OCD	No	200	No	13.6	No	64	93.8	206.4
15	Lithotripsy	260	OCM	No	150	10/12	13.6	13.6	45	54.4	163.1
16	Lithotripsy	50	OCM	No			6.8	No	6.8	No	26.4
17	Lithotripsy	550	Brushite	Yes	150	12/14	13.6	13.6	27.2	34	84.3
18	Lithotripsy	431	OCM	No	200	No	9.5	No	21.8	61.2	102
19	Lithotripsy	4000	OCD	No	150	10/12	9.5	8.1	34	61.2	119.6
20	Lithotripsy	658	OCD	Yes	150	10/12	15.0	17.6	27.2	34	69.3

OCD: oxalate calcium dehydrate. OCM: oxalate calcium monohydrate.

**Table 2 jcm-12-00147-t002:** Relationship between peak pressure, fluid absorption and postoperative complications.

Patient	Preoperative Urine Culture	Surgery Time (min)	Peak Pressure (cmH_2_O)	Fluid Balance (mL)	Postoperative Infection	Clavien–Dindo (<1 month)
1	Sterile	164	163	0	No	I
2	*S. agalactia* (Cefotaxime)	210	197	+300	Fever (4 days)	II
3	Sterile	145	176.7	+600	No	I
4	*E. coli* (Amoxicillin)	160	54.4	0	No	I
5	Sterile	120	30	0	No	I
6	Sterile	167	238	−100	No	O I
7	Sterile	102	15	−500	No	I
8	*P. aeruginosa* (Meropenem)	256	236.6	+650	No	I
9	Sterile	208	34	0	No	I
10	Sterile	147	171.3	−400	Outpatient	I
11	Sterile	185	99.2	−500	No	I
12	Sterile	117	156.3	+400	No	I
13	Sterile	135	122.4	0	Outpatient	I
14	Sterile	105	206.4	0	No	I
15	Sterile	199	163.1	+600	Fever (7 days)	II
16	Sterile	113	26.4	−200	Outpatient	I
17	*S. aureus* (Bactrim)	178	84.3	0	Fever (2 days)	II
18	Sterile	121	102	+500	Outpatient	I
19	*P. aeruginosa* (Tienam)	152	119.6	0	No	I
20	Sterile	60	69.3	0	No	I

## Data Availability

Data is unavailable due to privacy.

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
