# Peer review of "Real Time Intrarenal Pressure Control during Flexible Ureterorrenscopy Using a Vascular PressureWire: Pilot Study"

_jcm, 2022, doi:10.3390/jcm12010147_

Round 1

Reviewer 1 Report

In the article, the authors present their experience of using the PressureWire system to measure intrapelvic pressure during surgery. Based on their observations, they conclude that the proposed approach is simple, safe, and can be recommended. The main advantage of the approach is that continuous real-time PPI monitoring allows rapid detection of  high  pressure, minimizing the risk of surgical complications. The article is of particular interest to specialists and needs minor revision:

1) 62-71 Supplement section 2.2 with information about the sensor time constant, sampling rate and pressure measurement accuracy.

2) Since the accuracy of pressure measurement in the presented sensor does not exceed 2% (in [0.1007/s00345-020-03216-w] it is specified: Its accuracy is ± 1 mmHg plus ± 1% (≤ 50 mmHg) ± 3% (> 50 mmHg)), then all measured pressure values must not be presented with an excessive number of characters. For example, instead of (144) 6.8-47.58 you should write 6.8-48, instead of (148) 21.75, 63.90 you should write 21.8-64, etc. This applies to all the figures given, including those in Table 1, Table 2

Author Response

REVIEWER 1

The article is of particular interest to specialists and needs minor revision:

1) 62-71 Supplement section 2.2 with information about the sensor time constant, sampling rate and pressure measurement accuracy.

Pressure is measured every second. The pressure is measured in mmHg and the available range is from −30 to 300 mmHg (−40.8 to 407.9 cmH2O). Its accuracy is±1 mmHg plus±1% (≤50 mmHg)±3% (>50 mmHg).

This is being added in the manuscript, now reads: “Pressure is recorded every second. The pressure is measured in mmHg and the available range is from −30 to 300 mmHg (−40.8 to 407.9 cmH2O). Its accuracy is±1 mmHg plus±1% (≤50 mmHg)±3% (>50 mmHg). Pressure values measured in mmHg were multiplied by 1.35951 to convert them in cmH2O.”

2) Since the accuracy of pressure measurement in the presented sensor does not exceed 2% (in [0.1007/s00345-020-03216-w] it is specified: Its accuracy is ± 1 mmHg plus ± 1% (≤ 50 mmHg) ± 3% (> 50 mmHg)), then all measured pressure values must not be presented with an excessive number of characters. For example, instead of (144) 6.8-47.58 you should write 6.8-48, instead of (148) 21.75, 63.90 you should write 21.8-64, etc. This applies to all the figures given, including those in Table 1, Table 2

Thanks for the observation, we have simplified all pressure values that appear in the manuscript.

Reviewer 2 Report

Comment to the authors

The authors conducted a pilot study to evaluate the feasibility of IPP measurement with a PressureWire during flexible ureterorenoscopy. Total of 20 patients were recruited and maximum IPP during therapeutic period was 54.38-236.55 cmH2O.The authors concluded IPP monitoring with the PressureWire is reproducible, effective, and safe. I have a few comments as follows.

This manuscript is well-written and I like the concept of this methodology. However, prior to the current pilot study, the authors already published another pilot study with a smaller sample size to evaluate the feasibility of IPP monitoring with the PressureWire (World J Urol. 2021 Feb;39(2):555-561.). Since the feasibility of this methodology is already proved in the previous pilot study, I wonder if the current study as an additional pilot feasibility study is needed. I would like to see how IPP monitoring can contribute to the improved safety of flexible ureterorenoscopy with a larger sample size.

Methods

·         Could the authors clarify more detail of the study design? Is this retrospective or prospective study? From when were the patients recruited? Was the patient consecutively recruited or were there any patients removed from the analysis? Were there any aborted cases with this methodology?

Results

·         Although the physiological IPP ranges from 0 45 to 5 cmH2O, the median baseline IPP of the current study was 13.6 (IQR 6.8-47.58). Could the authors comment on the reason why the discordance exists?

·         Table 2. Could the authors clarify Clavien-Dindo grade of each post operative complication? Were there any other perioperative adverse event related to the PressureWire

Conclusions

·         Line 223. The authors concluded that the PressureWire method is safe and reproducible. However, I think the safety and reproducibility of this methodology have not been well-proven in the current study. I suggest that the authors would comment on this or modify the conclusion.

Minor

·          Figures 2 and 3 need alphabetical legends (a, b, c…) in accordance with figure legends.

Author Response

REVIEWER 2

Comment to the authors

The authors conducted a pilot study to evaluate the feasibility of IPP measurement with a PressureWire during flexible ureterorenoscopy. Total of 20 patients were recruited and maximum IPP during therapeutic period was 54.38-236.55 cmH2O.The authors concluded IPP monitoring with the PressureWire is reproducible, effective, and safe. I have a few comments as follows.

This manuscript is well-written and I like the concept of this methodology. However, prior to the current pilot study, the authors already published another pilot study with a smaller sample size to evaluate the feasibility of IPP monitoring with the PressureWire (World J Urol. 2021 Feb;39(2):555-561.). Since the feasibility of this methodology is already proved in the previous pilot study, I wonder if the current study as an additional pilot feasibility study is needed. I would like to see how IPP monitoring can contribute to the improved safety of flexible ureterorenoscopy with a larger sample size.

We appreciate your comments. In the previous pilot study, only 4 patients were included and all of them went through stone lasertripsy. This is clearly a larger series and we wander to include consecutive cases treating UTUC and UPJ not just urinary stones.

Methods

  • Could the authors clarify more detail of the study design? Is this retrospective or prospective study? From when were the patients recruited? Was the patient consecutively recruited or were there any patients removed from the analysis? Were there any aborted cases with this methodology?

A pilot study is by definition prospective. However, to clarify these statements now reads: “A prospective pilot study of consecutive patients undergoing f-URS for different treatments: kidney stone disease, pyelo-ureteral junction syndrome (UPJ) and diagnosis/treatment for upper tract urothelial carcinoma (UTUC) was performed between March to April 2022”. Any of the cases were aborted.

Results

  • Although the physiological IPP ranges from 0 45 to 5 cmH2O, the median baseline IPP of the current study was 13.6 (IQR 6.8-47.58). Could the authors comment on the reason why the discordance exists?

We have no physiological explanation. Since this device is digital, zeroing is quite simple and rapid, and it is done outside the patient. As we mention in the discussion (line 180), similar baseline IPP were also reported in literature. We placed the pressurewire during the cystoscopy and there should not be any retrograde black flow increasing the renal pressure. However, as we were treating mostly renal stones (median stone burden was 864 (50-9000) mm3) this might increase the physiological pressure inside the kidney.

  • Table 2. Could the authors clarify Clavien-Dindo grade of each post operative complication? Were there any other perioperative adverse event related to the PressureWire

There were no complications related to PressureWire placement and table 2 has been modified to clarify Clavien Dindo complications (<1month).

Conclusions

  • Line 223. The authors concluded that the PressureWire method is safe and reproducible. However, I think the safety and reproducibility of this methodology have not been well-proven in the current study. I suggest that the authors would comment on this or modify the conclusion.

We consider that PressureWire placement is safe as there were no complications related to PressureWire placement, in addition we suggest that by increasing the size it could work as a safety wire. Moreover, by following M&M steps its placement can be done easily and also as it is an intuitive device it is easy to obtain the data.

Minor

  • Figures 2 and 3 need alphabetical legends (a, b, c…) in accordance with figure legends.

Alphabetical legends are being added to figures 2 and 3.

Round 2

Reviewer 2 Report

The authors addressed the reviewer's comment appropriately.